# Strategies selection for building e-commerce platforms for agricultural wholesale markets: A tripartite evolutionary game perspective

Qianwen Luo[1]*, Yujie Wang[1], Yongtao Liu[2]

1 School of Business,Beijing Wuzi University,Tongzhou District,Beijing,China, 2 School of Finance and Taxation,Capital University of Economics and Business,Fengtai District,Beijing,China

* luoqwy@163.com

**Data Availability Statement:** All relevant data are within the manuscript and its Supporting information files.

## Abstract

The rapid advancement and widespread implementation of digital technology have created opportunities for the e-commerce transformation of agricultural wholesale markets. The building of e-commerce platforms in this process is of utmost importance and should be approached methodically. This article analyzes the interests and behavioral choices of the agricultural wholesale markets, local government, and wholesalers by establishing a tripartite evolutionary game model. It applies replicator dynamics equations to describe the evolutionary strategies of each party. The findings of the study indicate that the behavioral choices of agricultural wholesale markets, local government, and wholesalers are influenced by their initial intentions. Furthermore, there exists a degree of alignment between the choices made by agricultural wholesale markets and wholesalers. The building of e-commerce platforms by agricultural wholesale markets can be facilitated through direct and indirect government subsidies; this also motivates wholesalers to adopt and utilize these platforms. Agricultural wholesale markets may further incentivize wholesalers to utilize their own e-commerce platforms by offering additional benefits. On the other hand, if the agricultural wholesale markets demonstrate strong initial inclinations toward using third-party e-commerce platforms. In this scenario, the local government has the potential to promote the widespread use of these platforms by providing both direct and indirect financial incentives to these markets, as well as actively encouraging wholesalers to participate in them. This study presents policy recommendations for agricultural wholesale markets and local government to support the effective implementation of e-commerce platforms in the agricultural wholesaler markets and facilitate a smooth transition to e-commerce in agricultural wholesale markets.

## 1. Introduction

As conventional distribution firms, agricultural wholesale markets have proven essential in improving the connection between small producers and large markets;this is achieved through their architecture, rules, and mechanisms for price discovery [1]. In China, it has been

**Funding:** Beijing Municipal Social Science Foundation Project Grant, grant number 2020ZX028,Beijing Municipal Education Commission Scientific Research Programme Project Grant,grant number SM202310037004.

**Competing interests:** The authors have declared that no competing interests exist.

observed that agricultural wholesale markets have been observed to play a significant role in the distribution of products within rural areas, accounting for approximately 70% of the total distribution. Agricultural wholesale markets serve as a fundamental component of the fresh vegetable marketing system in Turkey [2]. Similarly, in Bulgaria, approximately 75% of fishermen participate in the direct or indirect sale of their products through agricultural wholesale markets [3]. The advent of e-commerce has led to the need for improved safety standards, transaction efficiency, responsiveness, information processing, and comprehensive services in agricultural wholesale markets due to the circulation of farm products. Therefore, in order to uphold the fundamental role of agricultural wholesale markets in agrarian circulation, it is imperative to undertake the restructuring of their business models. This endeavor is also crucial to industrial structural reform and advancement [4]. From this standpoint, the promotion of e-commerce transformation and the upgrading of agricultural wholesale markets has emerged as an inevitable trend in accordance with the progress of the era.

With the rapid development of digital technology, every industry is undergoing transformation and upgrading. In the agricultural field, digital innovation is one of the core elements of agricultural high-end equipment manufacturing system digitization and decarbonization strategy. The application of digital technology can help agricultural enterprises achieve more efficient equipment manufacturing and reduce carbon emissions [5]. In the construction industry, the development of digital technology contributes to improving the digital innovation performance of enterprises. The digital integration of the construction-integrated supply chain is a crucial factor affecting the digital innovation of construction industry enterprises [6]. In the manufacturing industry, the degree of application of digital technology and the intensity of investment in digital technology by enterprises are crucial factors in the digital transformation of manufacturing enterprises [7]. Through digital innovation, manufacturing enterprises can improve their competitiveness and profit levels and enhance their production processes and management efficiency by effectively utilizing resources, increasing production efficiency, and reducing costs [8]. This will lead to higher profits and a more stable market position [7]. Digital technology is also impacting agricultural wholesale markets, which are beginning to undergo e-commerce transformation. E-commerce platforms provide a channel of information exchange for all potential traders [9]. With the widespread use of digital technology, many wholesale markets have taken advantage of digital technology, establishing their own commodity trading platforms [10]. When constructing e-commerce platforms for agricultural wholesale markets, there are two alternatives available: developing self-built platforms or utilizing third-party e-commerce platforms. When agricultural wholesale markets opt for self-building, they have the opportunity to tailor and create platforms that align with their specific requirements and distinctive attributes. The platforms have the capability to be created and tailored to meet the specific needs of market participants, offering a wide range of comprehensive features and services. Self-built platforms also give enhanced control over brand perception and user interaction, facilitating the provision of tailored services and solutions. Nevertheless, the development of self-built platforms necessitates a significant allocation of financial resources and technical expertise for their development and management. Additionally, these platforms may encounter difficulties in attracting a sufficient number of users during their early phase. By opting to utilize third-party platforms, agricultural wholesale markets can expedite the process of selling and promoting their products by seamlessly integrating into pre-existing platforms. Moreover, third-party platforms typically offer sophisticated logistics and payment systems, thereby streamlining the supply chain management processes within agricultural wholesale markets. However, the utilization of third-party platforms may result in a lack of complete control over brand image and user experience. Additionally, the presence of advertisements and competing brands on these platforms might potentially influence sales

outcomes. Hence, in the context of the e-commerce revolution within agricultural wholesale markets, the selection of suitable approaches for the systematic and rational build of e-commerce platforms emerges as a critical matter necessitating meticulous deliberation. To effectively tackle this issue, the present study formulates a three-party evolutionary game model that includes agricultural wholesale markets, local government, and wholesalers. It investigates the evolutionary dynamics of equilibrium strategies within the system under varying conditions and subsequently assesses the influence of relevant factors on system stability through numerical simulations. This study aims to answer the following questions.

1. How can the promotion of self-built e-commerce platforms in agricultural wholesale markets be enhanced, and what strategies may be employed to incentivize wholesalers to choose these platforms?

2. What strategies can be used to improve the promotion of third-party e-commerce platforms in agricultural wholesale markets and foster increased adoption among wholesalers?

This research study provides the following contributions: (1) This study aims to examine the process of e-commerce transformation in agricultural wholesale markets, with a specific focus on the crucial task of constructing e-commerce platforms. (2) This study aims to provide recommendations and insights for developing e-commerce platforms in agricultural wholesale markets by analyzing the options of self-built e-commerce platforms and utilizing third-party e-commerce platforms. (3) This study examines the impact of government subsidies on developing e-commerce platforms in agricultural wholesale markets. It aims to provide a scientific basis for the government to effectively implement policy measures that promote the construction of e-commerce platforms in agricultural wholesale markets. (4) This study examines the behavioral decisions made by agricultural wholesale markets, local government, and wholesalers, considering various original intents. (5) This study investigates the reciprocal impact of behavioral decisions and provides references and recommendations for developing e-commerce platforms in agricultural wholesale markets. The rest of this paper is structured as follows: The second part reviews existing research. The third part introduces the construction process of a tripartite evolutionary game model involving agricultural wholesale markets, local government, and wholesalers. The fourth part solves the game mode, explicitly examining the equilibrium strategies of each entity involved in constructing e-commerce platforms in agricultural wholesale markets. The fifth part conducts numerical simulations to analyze the behavioral choices of the three parties under different initial intentions and examines the key factors that influence their behavioral choices. The sixth and seventh parts summarize the research, offering pertinent suggestions for the scientific and rational construction of e-commerce platforms in agricultural wholesale markets and elaborating on this study's limitations and future research directions.

## 2. Literature review

The literature on this topic mainly focuses on the following areas: (1) research on the e-commerce transformation of agricultural wholesale markets. (2) research on evolutionary game theory.

### 2.1 Research on the e-commerce transformation of agricultural wholesale markets

The e-commerce transformation of agricultural wholesale markets is a process of technological advancements. Technological advancements can significantly reduce production costs,

improving productivity with fewer inputs [11]. E-commerce is also a shining manifestation of technological progress in the Internet era and is gradually permeating all aspects of social development [12]. Song et al. believed that information and communication technology have brought opportunities to traditional agricultural wholesale markets [13]. E-commerce technology has brought competitive advantages to conventional agricultural markets by helping them adjust their business processes [14] and reducing potential risks in the farm supply process [15]. Tan et al. argued that e-commerce technology can improve operational efficiency and traceability in agricultural products [16]. Chaudhary et al. claimed that e-commerce in agricultural wholesale markets will be more open and transparent than physical markets [17]. Additionally, e-commerce technology has the potential to decrease high transaction costs and search costs resulting from incomplete or asymmetric information [18] while also assisting e-commerce platform users in increasing their income [19]. Joshi et al. argued that developing e-commerce technology is essential for developing countries to address food safety during emergency outbreaks [20]. Kosior et al. believed that e-commerce technology can improve the competitiveness of agricultural wholesale markets and enable them to utilize resources more effectively [21]. However, the adoption of e-commerce technology also brings challenges to agricultural wholesale markets [12]. The adoption of e-commerce technology also poses challenges for agricultural wholesale markets, as it increases transaction costs and triggers competition from influential e-commerce channels, which may necessitate traders to readjust or reconfigure their strategies [22].

While transforming agricultural wholesale markets into e-commerce, Zengjun reviewed the development history of agricultural wholesale markets. He believed that since the establishment of the People's Republic of China, agricultural wholesale markets have undergone spontaneous emergence, rapid development, blind development, standardized development, and qualitative improvement. With the application and widespread use of digital technology, the intelligent "third-generation agricultural wholesale market" is the future direction of development [23]. Kamble et al. believed that technological progress drives the continuous development of agricultural wholesale markets. In transitioning traditional agricultural wholesale markets towards e-commerce-driven platforms, it is crucial for practitioners to make scientific and reasonable investments [24]. Reardon et al. suggested strengthening investments in the infrastructure of agricultural wholesale markets and reducing policy restrictions by simplifying regulations [25]. Yadav et al. emphasized that support from skilled technical personnel is critical for promoting the digitization of agricultural supply chains. Meanwhile, infrastructure and logistics play crucial roles [26]. Vern et al. argued that in the process of e-commerce transformation, well-trained labor, abundant funding, and preliminary investigations are crucial factors [27]. Providing timely government subsidies in this process is a more practical approach [28], as these subsidies significantly improve business performance [29]. As a lever for market regulation and resource allocation, the government provides generous financial support to enterprises, aiming to widely replace outdated technologies in traditional industries [30]. The government also plays a significant role in promoting economic development, technological innovation, and product innovation [31]. Zeng believed that digitizing infrastructure can enhance the management level of agricultural wholesale markets. Additionally, the government can enhance the innovative capacity of agricultural wholesale markets through supportive policies [32].

## 2.2 Evolutionary game models and their applications

In recent years, the development of game theory has led to its wide application in various research fields. Traditional game theory assumes that decision-makers are "completely rational." Still, in reality, decision-makers are limited by their own cognition and environmental

information differences and can only make decisions that conform to expectations rather than being optimal [33]. Therefore, game theory has gradually combined with other disciplines, enriching and developing game theory. The combination of game theory and quantum information science has formed quantum game theory. Yu et al. applied quantum game theory in the development of rural new energy, explaining the entanglement mechanism between new energy enterprises and rural collectives and establishing incentive mechanisms [34]. The decision-making process of decision-makers is often influenced by various stochastic interference factors, leading to significant uncertainties in the equilibrium results. Therefore, some scholars use the differential game theory to explore the issues. Yin et al. applied differential game theory to analyze the issue of sharing low-carbon technology between dominant and inferior companies [35]. In the game process, the behavior adjustment of game participants should be regarded as a dynamic evolution process, which is the core where evolutionary game theory breaks the assumption of complete rationality [36]. Evolutionary game theory emphasizes dynamic equilibrium, which can demonstrate a group's dynamic evolution process over time, thus finding optimal strategies and resolving the issue of actors' incomplete rationality [37]. It has been widely applied in supply chain management and system structural evolution research [38]. Tang et al. proposed that when companies conduct e-commerce transformation and upgrading, they need to ensure their own profits and good business operations by constructing an evolutionary game model among companies, government, and digital technology platforms [39]. Dong et al. used an evolutionary game model to discover that the government's subsidy intensity affects the digital transformation process of enterprises and put forward suggestions such as strengthening digital infrastructure construction, establishing a subsidy dynamic adjustment mechanism, and constructing a collaborative and synergetic digital ecosystem [40]. Hou et al. also believed that excessive government intervention would affect market operations [41]. Sun et al. utilized dual-population evolutionary game theory to investigate the evolutionary process and influencing factors of the cooperation mechanism in the pork supply chain and devise a series of measures to promote effective competition and cooperation within the high-quality pork supply chain. These measures include reducing cooperation costs, increasing cooperation income, and establishing a fair and efficient system for income distribution and risk compensation [42]. Through the analysis of an evolutionary game model, Lin et al. discovered that the benefits of digital technology empowerment, spillover effects, and supervisory effects exhibit positive correlations with governments, platform centers, and node enterprises' willingness to undergo e-commerce transformation [43].

## 2.3 Summary

Presently, experts from both domestic and international contexts have unequivocally acknowledged the crucial significance of e-commerce technology in facilitating the transformative process of agricultural wholesale markets within the realm of electronic commerce. The facilitation of e-commerce transformation in agricultural wholesale markets is achievable through various means, such as improving infrastructure and implementing supportive government policies. However, it is imperative to conduct a comprehensive study to ascertain the scientific methods for building e-commerce platforms during the process of transforming agricultural wholesale markets into e-commerce. Furthermore, the majority of existing research primarily focuses on examining the use of game theory in analyzing the process of e-commerce transformation undertaken by firms. In contrast, there is limited research on firms' scientific build of e-commerce platforms. Hence, this study employs evolutionary game theory as a framework to examine the trajectory of agricultural wholesale markets in adopting e-commerce platforms, taking into account various initial intentions. Additionally, this research

offers relevant recommendations to facilitate the transformation of agricultural wholesale markets into e-commerce entities.

## 3. Description and build of evolutionary game model

### 3.1 Model assumptions and parameter settings

Assumption 1: Game players

The process of building e-commerce platforms for agricultural wholesale markets involves dynamic interactions among multiple parties. The promotion of e-commerce platforms in agricultural wholesale markets can be facilitated by the local government through the provision of subsidies. Consequently, the behavioral choices made by wholesalers have a direct impact on the behavioral choices observed within the agricultural wholesale markets. The construction of e-commerce platforms in agricultural wholesale markets is greatly facilitated by the co-creation value between markets and wholesalers [44]. Hence, this research endeavors to establish a trilateral evolutionary game model encompassing agricultural wholesale markets, local government, and wholesalers. The game in question falls under the category of a non-symmetric information dynamic game. All participants involved in the game exhibit bounded rationality as they seek to optimize their own interests and adapt their methods iteratively throughout the game.

Assumption 2: Strategy space

The building of e-commerce platforms in agricultural wholesale markets necessitates the involvement of three key stakeholders: agricultural wholesale markets, local government, and wholesalers. The agricultural wholesale markets are considering the potential implementation of e-commerce platforms. The strategic options available for agricultural wholesale markets include building self-built e-commerce platforms and using third-party e-commerce platforms. The probability of agricultural wholesale markets choosing to build their own e-commerce platforms is denoted as $x$, while the probability of using third-party e-commerce platforms is represented as $1 - x (0 \leq x \leq 1)$. The local government is currently deliberating on the potential provision of subsidies for constructing e-commerce platforms within agricultural wholesale markets. The strategic options available to local government include providing subsidies or refraining from providing subsidies. The variable $y$ represents the probability that local government will choose to offer subsidies, while the probability of not providing subsidies is represented by $1 - y (0 \leq y \leq 1)$. The strategic space available to wholesalers encompasses using self-built e-commerce platforms within agricultural wholesale markets or opting for third-party e-commerce platforms. The probability that wholesalers choose to use self-built e-commerce platforms of agricultural wholesale markets is denoted as $z$, while the probability of choosing third-party e-commerce platforms is represented as $1 - z (0 \leq z \leq 1)$.

Assumption 3: Benefits to stakeholders and corresponding costs

When agricultural wholesale markets want to build their own e-commerce platforms, they commonly encounter charges related to technology development, maintenance, human resource expenditures, and other associated costs designated as $C_l$. However, when agricultural wholesale markets want to utilize third-party e-commerce platforms, they typically incur expenses associated with the management and operation of e-commerce enterprises, referred to as $C_t$. Subsidies are of paramount importance in incentivizing the building of e-commerce platforms for agricultural wholesale markets by local government. Local government subsidies can be categorized into two main types: direct subsidies and indirect subsidies, which may include tax incentives [45]. When the local government chooses to allocate subsidies, both self-built e-commerce platforms and third-party platforms utilized by agricultural wholesale markets will be eligible to receive direct and indirect financial assistance from the local government. The local government designates the direct subsidy amount as variable $I$, while the

allocation coefficient is represented by variable $U$. When agricultural wholesale markets choose to build their own e-commerce platforms, they usually encounter expenses related to technology development and maintenance and costs associated with human resources. Conversely, agricultural wholesale markets that decide to utilize third-party platforms will receive direct subsidies of $(1 - u)I$ from the local government. When agricultural wholesale markets choose to build their own e-commerce platforms, the cost that the markets have to bear is represented by $m$. This means that the indirect subsidy provided by the local government for the agricultural wholesale markets to build their own e-commerce platforms is $1 - m$. On the other hand, when agricultural wholesale markets choose to use third-party e-commerce platforms, the cost that the markets have to bear is $1 - m$, and the indirect subsidy provided by the local government for the agricultural wholesale markets using third-party platforms is $m$. Local government can promote the digital transformation process of agricultural wholesale markets through subsidies. The central government recognizes the importance of these measures and encourages and supports the efforts of local government through rewards. Through this mechanism, the central government aims to achieve balanced development nationwide, promote economic prosperity in different regions, and strengthen cooperation and interaction between local and central government. Therefore, if the local government chooses to provide subsidies, it will receive rewards from the central government, referred to as $A$ [46]. Assuming that the wholesalers choose to utilize the internally developed electronic commerce systems offered by the agricultural wholesale markets. In such circumstances, wholesalers must compensate the platform for its use and pay commissions to the agricultural wholesale markets, referred to as $T_1$. In the event that wholesalers choose to use third-party e-commerce platforms, they are required to remunerate the platform provider, referred to as $T_2$ by paying usage fees and commissions. When wholesalers want to utilize the self-built e-commerce platforms offered by agricultural wholesale markets, they might generate additional revenue, referred to as $U_p$, due to the enhanced provision of personalized services by these self-built platforms. When the platform model utilized in agricultural wholesale markets aligns with the platform model favored by the wholesalers, both parties can attain synergistic advantages, denoted as $b$. However, if the platforms used by agricultural wholesale markets do not align with the preferences of wholesalers, it may be necessary for these markets to attract a broader customer base. The potential consequences of this situation may have adverse effects on the progress of regional economic development, primarily stemming from the hindered ability to facilitate the transition of agricultural wholesale markets towards e-commerce effectively. The loss incurred by the local government is represented by the variable $w_2$, while the loss of customer sources for the agricultural wholesale markets is represented by the variable $w_1$. The parameter settings are presented in Table 1.

### 3.2 Game payoff matrix

The payoff matrix for the game model based on the above parameters and assumptions is shown in Table 2.

## 4. Model solving

### 4.1 Trilateral replication dynamic analysis

According to the payment matrix, the expected revenue of choosing to build e-commerce platforms for agricultural wholesale markets is $U_1$, thus:

$$U_1 = yz(uI - mC_l + T_1 + b) + y(1 - z)(uI - w_2 - mC_l) + (1 - y)z(-C_l + b + T_1) \\ + (1 - y)(1 - z)(-w_2 - C_l) \tag{1}$$

**Table 1. Description of economic parameter variables in the game model.**

| Stakeholder | Parameter | Parameter description |
|---|---|---|
| agricultural wholesale markets | $C_l$ | Cost of building e-commerce platforms for agricultural wholesale markets |
| | $C_t$ | Cost of using third-party platforms for agricultural wholesale markets |
| | b | Both agricultural wholesale markets and wholesalers can benefit synergistically when their behaviors align |
| | $w_2$ | Loss of customer sources for agricultural wholesale markets when their behaviors are not aligned with wholesalers |
| local government | I | Amount of subsidy when agricultural wholesale markets choose to provide subsidies |
| | m | The local government indirectly subsidizes agricultural wholesale markets for using third-party e-commerce platforms. |
| | A | Rewards are given by the central government when local government chooses to provide subsidies |
| | $w_1$ | Loss incurred by local government when the behaviors of agricultural wholesale markets and wholesalers are not aligned |
| | u | The proportion of direct subsidies the local government gives agricultural wholesale markets for building their own e-commerce platforms |
| | 1-m | The local government indirectly subsidizes agricultural wholesale markets for building their own e-commerce platforms |
| | 1-u | The proportion of direct subsidies the local government gives to agricultural wholesale markets for using third-party e-commerce platforms |
| wholesalers | $T_1$ | Wholesalers must pay fees when using agricultural wholesale markets' self-built e-commerce platforms |
| | $T_2$ | Fees that wholesalers have to pay when using third-party e-commerce platforms |
| | Up | Wholesalers can obtain additional benefits when using self-built e-commerce platforms of agricultural wholesale markets |

**Table 2. Game payoff matrix.**

| Participants of the evolutionary game | | Wholesalers | |
|---|---|---|---|
| | | **Using the e-commerce platforms of agricultural wholesale markets z** | **Using third-party e-commerce platforms 1-z** |
| Agricultural wholesale markets are building their own e-commerce platforms x | Local government provides subsidies y | $uI + mC_l + T_1 + b$ | $uI - w_2 - mC_l$ |
| | | A- uI<br>$-T_1 + U_p + b$ | $-w_1 + A - uI$<br>$-T_2$ |
| | Local government does not provide subsidies 1-y | $-C_l + b + T_1$ | $-w_2 - C_l$ |
| | | 0<br>$-T_1 + b + U_p$ | $-w_1$<br>$-T_2$ |
| Agricultural wholesale markets using third-party e-commerce platforms 1-x | Local government provides subsidies y | $-w_2 + (1-u)I - (1-m)C_t$ | $b+(1-u)$ I- $(1-m)C_t$ |
| | | A-$w_1 - (1-u)I$<br>$-T_1 + U_p$ | A-(1-u) I<br>b-$T_2$ |
| | Local government does not provide subsidies 1-y | $-w$ 2 $- C_t$ | b-$C_t$ |
| | | $-w_1$<br>$-T_1 + U_p$ | 0<br>b-$T_2$ |
| From top to bottom: agricultural wholesale markets, local government, wholesalers | | | |

The expected revenue of choosing to use third-party e-commerce platforms for agricultural wholesale markets is $U_2$, thus:

$$U_2 = yz(-w_2 + (1-u)I - (1-m)C_t) + y(1-z)(b + (1-u)I - (1-m)C_t) \\ + z(1-y)(-w_2 - C_t) + (1-y)(1-z)(b - C_t) \tag{2}$$

The replication dynamic equation for agricultural wholesale markets is as follows:

$$F(x) = dx/dt = x(U_1 - U_x) = x(1-x)(zT_1 + 2bz + 2w_2z + 2Iuy - C_tmy - \\ C_tmy + C_ty - C_l + C_t - b - w_2 - Iy) \tag{3}$$

According to the payment matrix, the expected revenue of the local government's choice to provide subsidies is $U_3$, thus:

$$U_3 = xz(A - uI) + x(1-z)(-w_1 + A - uI) + (1-x)z(A - w_1 - (1-u)I) \\ + (1-x)(1-z)(A - (1-u)I) \tag{4}$$

The expected revenue of the local government's choice not to provide subsidies is $U_4$, thus:

$$U_4 = x(1-z)(-w_1) + (1-x)z(-w_1) \tag{5}$$

The replication dynamic equation for the local government is:

$$F(y) = dy/dt = y(U_3 - U_y) = y(1-y)(A - I + Iu + Ix - 2Iux) \tag{6}$$

According to the payment matrix, the expected profit of the wholesalers using the e-commerce platforms of agricultural wholesale markets is $U_5$, thus:

$$U_5 = xy(-T_1 + U_p + b) + x(1-y)(-T_1 + b + U_p) + (1-x)y(U_p - T_1) \\ + (1-x)(1-y)(U_p - T_1) \tag{7}$$

The expected profit of the wholesalers using the third-party e-commerce platforms is $U_6$, thus:

$$U_6 = -xyT_2 + x(1-y)(-T_2) + (1-x)y(b - T_2) + (1-x)(1-y)(b - T_2) \tag{8}$$

The replication dynamic equation of wholesalers is:

$$F(z) = dz/dt = z(U_5 - U_z) = z(1-z)(T_2 - T_1 + U_p - b + 2bx) \tag{9}$$

## 4.2 Analysis of strategy stability under co-evolutionary dynamics

The asymptotic stability of an equilibrium point can be determined by constructing the Jacobian matrix and solving for the eigenvalues. The Jacobian matrix can be obtained by taking the partial derivatives of the replicator dynamic equations F(x), F(y), and F(z) with respect to x, y, and z, respectively, as shown below:

$$J = \begin{Bmatrix} \dfrac{\delta F(x)}{\delta x} & \dfrac{\delta F(x)}{\delta y} & \dfrac{\delta F(x)}{\delta z} \\ \dfrac{\delta F(y)}{\delta x} & \dfrac{\delta F(y)}{\delta y} & \dfrac{\delta F(y)}{\delta z} \\ \dfrac{\delta F(z)}{\delta x} & \dfrac{\delta F(z)}{\delta y} & \dfrac{\delta F(z)}{\delta z} \end{Bmatrix} = \begin{Bmatrix} F_{11} & F_{12} & F_{13} \\ F_{21} & F_{22} & F_{23} \\ F_{31} & F_{32} & F_{33} \end{Bmatrix}$$

In this context:

$$F_{11} = (1 - 2x)(T_1 z + 2bz + 2w_2 z + 2Iuy - C_l my - C_t my + C_l y - b - w_2 + C_t - C_l);$$
$$F_{12} = x(1 - x)(2Iu - C_l mI - C_t m + C_l - I); F_{13} = x(1 - x)(T_1 + 2b + 2w_2)$$
$$F_{21} = y(1 - y)(I - 2Iu); F_{22} = (1 - 2y)(A - I + Iu + Ix - 2Iux); F_{23} = 0$$
$$F_{31} = z(1 - z)(2b); F_{32} = 0; F_{33} = (1 - 2z)(-T_1 + T_2 + U_p - b + 2bx)$$

When each equilibrium point is stable, the replicator dynamic equations for agricultural wholesale markets, local government, and wholesalers must equal zero.

$$\begin{cases} dx/dt = 0 \\ dy/dt = 0 \\ dz/dt = 0 \end{cases}$$

The pure strategy equilibria are the evolutionarily stable points in asymmetric games caused by information asymmetry. Therefore, based on the replicator dynamic equations, only eight pure strategy equilibrium points are discussed: (0,0,0), (0,0,1), (0,1,0), (0,1,1), (1,0,0), (1,0,1), (1,1,0), and (1,1,1). According to Lyapunov's theorem, when all eigenvalues of the Jacobian matrix are "-," the corresponding points are the equilibrium points of the evolutionary game. The eigenvalues for various equilibrium points are shown in Table 3.

Scenario 1: When the value of $A - I - I \cdot u > 0$, the local government will choose to provide subsidies. Given the conditions $2I \cdot u - C_l \cdot m - C_t \cdot m - b - w2 - I + Ct < 0$ and $T_2 - T_1 + U_p - b < 0$, it can be determined that the equilibrium point of the system is located at coordinates (0, 1, 0). The corresponding evolutionarily stable strategy is (agricultural wholesale markets using third-party e-commerce platforms, local government provides subsidies, and wholesalers using third-party e-commerce platforms). In this case, the local government chooses to provide subsidies, and there is consistency in the behavior choices of agricultural wholesale markets and wholesalers. The agricultural wholesale markets choose to use third-party e-commerce platforms, and wholesalers also use third-party e-commerce platforms.

Scenario 2: When the value of $A - I \cdot u > 0$, the local government will opt to provide subsidies. Given the conditions $T_1 + b + w_2 + 2I \cdot u - m \cdot C_l - m \cdot C_t - I + C_t > 0$ and $T_2 - T_1 + U_p + b > 0$, it can be concluded that the equilibrium point of the system is located at coordinates (1, 1, 1), and the choices of all parties are (agricultural wholesale markets building their own e-commerce platforms, local government provides subsidies, and wholesalers choosing to use the agricultural wholesale markets' e-commerce platforms). In this case, the local government will choose to provide subsidies, and there is consistency in the behavior choices of agricultural

**Table 3. Stability analysis of equilibrium points.**

| | $\lambda_1$ | $\lambda_2$ | $\lambda_3$ | Stability |
|---|---|---|---|---|
| (0,0,0) | $-C_l + C_t - b - w_2$ | A-I+Iu | $T_2 - T_1 + U_p - b$ | Unstable point |
| (0,0,1) | $T_2 + w_2 + b - C_l + C_t$ | A-I+Iu | $-(T_2 - T_1 + U_p - b)$ | Unstable point |
| (0,1,0) | $2Iu - C_l m - C_t m - b - w_2 - I + C_t$ | $-(A-I+Iu)$ | $T_2 - T_1 + U_p - b$ | Conditionally stable ESS |
| (0,1,1) | $T_1 + b + w_2 + 2Iu - C_l m - C_t m - I + C_t$ | $-(A-I+Iu)$ | $-(T_2 - T_1 + U_p - b)$ | Unstable point |
| (1,0,0) | $-(-C_l + C_t - b - w_2)$ | A-Iu | $T_2 - T_1 + U_p + b$ | Unstable point |
| (1,0,1) | $-(T_1 + b + w_2 - C_l + C_t)$ | A-Iu | $-(T_2 - T_1 + U_p + b)$ | Unstable point |
| (1,1,0) | $-(2Iu - mC_l - mC_t - b - w_2 - I + C_t)$ | $-(A-Iu)$ | $T_2 - T_1 + U_p + b$ | Unstable point |
| (1,1,1) | $-(T_1 + b + w_2 + 2Iu - mC_l - mC_t - I + C_t)$ | $-(A-Iu)$ | $-(T_2 - T_1 + U_p + b)$ | Conditionally stable ESS |

wholesale markets and wholesalers. The agricultural wholesale markets tend to choose to build their own e-commerce platforms, and wholesalers also tend to use the agricultural wholesale markets' self-built e-commerce platforms.

## 5. Simulation analysis

The present work utilizes MATLAB as a tool for conducting numerical simulation and analysis. The validation of the usefulness and accuracy of the evolutionary stability analysis is achieved by adding specific parameter values and conducting intuitive demonstrations. This study examines the strategic decisions made by agricultural wholesale markets in developing e-commerce platforms. Specifically, it investigates the impact of initial preferences of the three-party evolutionary game players and essential parameters in the replicator dynamics equations on the evolutionary behavior strategies of agricultural wholesale markets, local government, and wholesalers. This analysis aids in identifying the most advantageous approach for constructing e-commerce platforms within agricultural wholesale markets.

### 5.1 Basic situation simulation

While the initial values of the parameters do have some impact on the rate and scope of their evolution, they do not alter the overall patterns and outcomes. The accuracy and practicality of the model are essential considerations in the process of establishing a model. This article obtains the initial values of various parameters through four methods. The first method is from policies. Since the issuance of the "Opinions on Accelerating the Development of the Circulation Industry" by the State Administration for Industry and Commerce of China in 2013, policies have been introduced across the country to support the development of e-commerce in agricultural wholesale markets after the establishment of online sales platforms. In Foshan, Guangdong, the government provides incentives of 0.4 million and 0.2 million yuan respectively for the construction of five-star and four-star smart agricultural wholesale markets. In Dongguan, Guangdong, the government provides a subsidy of 30% of the investment for the construction of smart agricultural wholesale markets, with a maximum of 0.5 million yuan. In Shishi, Fujian, and Haining, Zhejiang, the local finance department provides subsidies of up to 0.1 million yuan for the transformation of agricultural wholesale markets into e-commerce. In Qingdao, Shandong, the municipal government provides an incentive of up to 0.3 million yuan for the transformation into e-commerce of agricultural wholesale markets. In Jiaozuo, Henan, and Huaihua, Hunan, the municipal government provides financial support of up to 0.5 million yuan for the construction of smart agricultural wholesale markets. The implementation of these policies has stimulated the transformation of agricultural wholesale markets into e-commerce. Based on policies across the country, it is assumed that $I$ = 30 (Units: ten thousand yuan). The second method is based on existing literature. Referring to the research by Cui et al. [47] on platform enterprises, government, and consumers, $T_1$ = 15 and $T_2$ = 10 (Units: ten thousand yuan) are assumed. Referring to the research by Li et al. [46] on the promotion of regional public brands for agricultural products and the analysis by Guo et al. [48] on the construction of agricultural product brands, $A$ = 30 and $U_p$ = 5(Units: ten thousand yuan) are assumed. $B$ = 30 (Units: ten thousand yuan) is assumed based on the research by Guo et al. [48] on the construction of agricultural product brands, and $W_2$ = 20 (Units: ten thousand yuan) is assumed based on the research by Sara et al. [49] on the adoption of electric vehicles and the research by Guo et al. [48] on the construction of agricultural product brands. The third method is based on field research. This article assumes that the incentives from the government for agricultural wholesale markets to build their own e-commerce platforms and use third-party platforms are the same at the initial stage, that is, u = 0.5 and m = 0.5. At the

same time, the influence of different levels of government support on the behavior of agricultural wholesale markets and wholesalers is also discussed in the subsequent discussions. The fourth method is based on interviews with managers of agricultural wholesale markets and data collection. This article obtained that the cost for agricultural wholesale markets to choose to build their own e-commerce platforms is approximately 0.4 million to 0.5 million yuan through interviews with managers of agricultural wholesale markets. By reviewing the fees for joining third-party platforms, it is found that the fees mainly include platform service fees, margin, promotion fees, and technical service fees. The platform service fees are about 0.05 million to 0.2 million yuan, the margin is about 0.05 million to 0.1 million yuan, the promotion fees are about 0.03 million to 0.06 million yuan, and the technical service fees, which are commission, are about 5% to 12% of the sales revenue. Therefore, it is assumed that $C_l = 45$ and $C_t = 35$(Units: ten thousand yuan). Based on the above analysis, the assigned values for each parameter are as follows: $b = 30; C_l = 45; C_t = 35; w_2 = 20; I = 30; T_1 = 15; T_2 = 10; A = 30; U_p = 5; u = 0.5; m = 0.5$(Units: ten thousand yuan). To verify the accuracy of the parameter assignments, this study conducted simulations under basic conditions and obtained the simulation results as seen in Fig 1.

From Fig 1, it can be observed that the system has two equilibrium points at this time, namely (0, 1, 0) and (1, 1, 1), which represent the behavioral choices of each party as follows: the first equilibrium point illustrates the choices of the agricultural wholesale markets using third-party platforms, local government provedes subsidies, and wholesalers using third-party e-commerce platforms. The second equilibrium point represents the agricultural wholesale markets' choices to build their own e-commerce platforms, local government provides subsidies, and wholesalers using the agricultural wholesale markets' e-commerce platforms. These choices are consistent with the theoretical analysis mentioned earlier. Therefore, this parameter assignment has a certain level of validity.

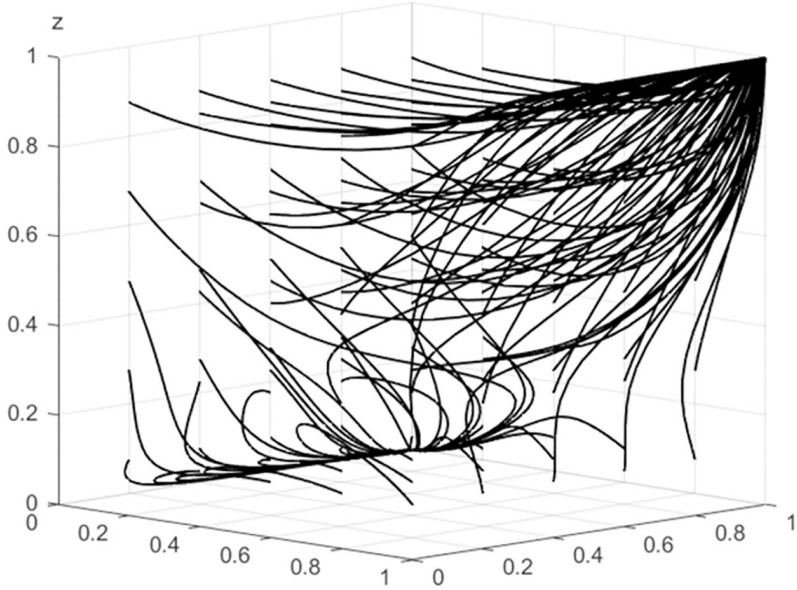

**Fig 1. Results after 50 iterations of evolution.**

## 5.2 The influence of changes in initial intentions of subjects on system evolution

Referring to the methods of parameter setting by Liu et al. [43] and Liu et al. [50], the initial intentions of agricultural wholesale markets to choose self-built e-commerce platforms, government provedes subsidies, and wholesalers to choose to use self-built e-commerce platforms of agricultural wholesale markets were set at three levels: low, medium, and high, denoted as x, y, z ∈ Ω(0.2, 0.5, 0.8). The results of the behavioral evolution strategy are shown as follows.

**5.2.1 The impact of changes in initial intentions of subjects on the evolution of behavioral strategies in agricultural wholesale markets.** Changes in the initial intentions of agents have varying degrees of impact on the evolution of behavioral strategies in agricultural wholesale markets. As shown in Fig 2, when the initial intention of agricultural wholesale markets to use self-built e-commerce platforms is low, that is, the initial intention to use third-party e-commerce platforms is high ($x_0 = 0.2$), if the initial intention of local government subsidies is also low ($y_0 = 0.2$), the agricultural wholesale markets will choose to use third-party e-commerce platforms. As the initial intention of local government subsidies gradually increases, the behavioral choices of agricultural wholesale markets will shift from using third-party e-commerce platforms to building e-commerce platforms, and the speed will increase progressively. When the initial intention of agricultural wholesale markets to use self-built e-commerce platforms is high ($x_0 \geq 0.5$), the initial intention of local government subsidies has no impact on the behavioral choice of agricultural wholesale markets to build their own e-commerce platforms. However, the speed of building their own e-commerce platforms increases with the continuous increase in the initial intention of local government subsidies. As shown in Fig 3, when the initial intention of agricultural wholesale markets to use third-party platforms is high, indicating a low initial intention to build their own e-commerce platforms ($x_0 =$

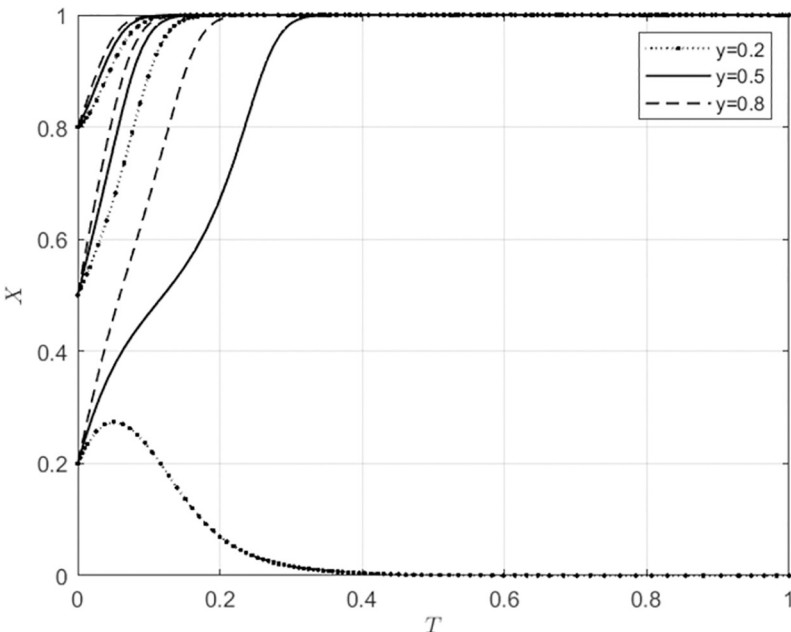

**Fig 2. Evolutionary strategies of agricultural wholesale markets behavior under different initial intentions of local government.**

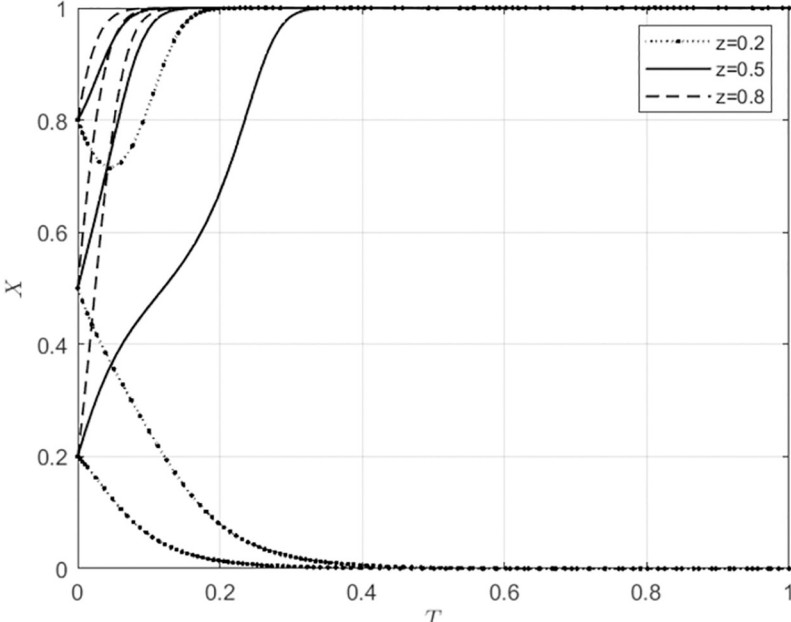

**Fig 3. Evolutionary strategies of agricultural wholesale markets behavior under different initial intentions of wholesalers.**

0.2), if the initial intention of wholesalers to use the self-built e-commerce platforms of the agricultural wholesale markets is low ($z_0 = 0.2$), the agricultural wholesale markets will choose to use third-party e-commerce platforms. As the initial intention of wholesalers to use self-built e-commerce platforms of the agricultural wholesale markets gradually increases, the behavioral choices of the agricultural wholesale markets will shift from using third-party e-commerce platforms to building their own e-commerce platforms at an increasing speed. When the initial intention of agricultural wholesale markets to build their own e-commerce platforms is at a moderate level ($x_0 = 0.5$), if the initial intention of wholesalers to use the self-built e-commerce platforms of the agricultural wholesale markets is low ($z_0 = 0.2$), the agricultural wholesale markets will choose to use third-party e-commerce platforms. As the initial intention of wholesalers to use self-built e-commerce platforms of the agricultural wholesale markets gradually increases, the behavioral choices of the agricultural wholesale markets will shift from using third-party e-commerce platforms to building their own e-commerce platforms, and the speed of choosing to build their own e-commerce platforms will gradually increase. When the initial intention of agricultural wholesale markets to build their own e-commerce platforms is at a high level ($x_0 = 0.8$), as the initial intention of wholesalers to use the self-built e-commerce platforms of the agricultural wholesale markets gradually increases, the speed of building their own e-commerce platforms will gradually increase. This indicates that the initial intentions of local government and wholesalers influence the behavioral choices of agricultural wholesale markets.

**5.2.2 The impact of initial intention changes of subjects on the evolutionary strategies of local government behavior.** From Figs 4 and 5, the initial intentions of self-built e-commerce platforms for agricultural wholesale markets and the choices of wholesalers to use these platforms will not affect the evolutionary direction of local government behavior strategies. Local government will always choose to provide subsidies, and the speed of their subsidy selection will increase as their initial willingness to subsidize gradually increases. This indicates that

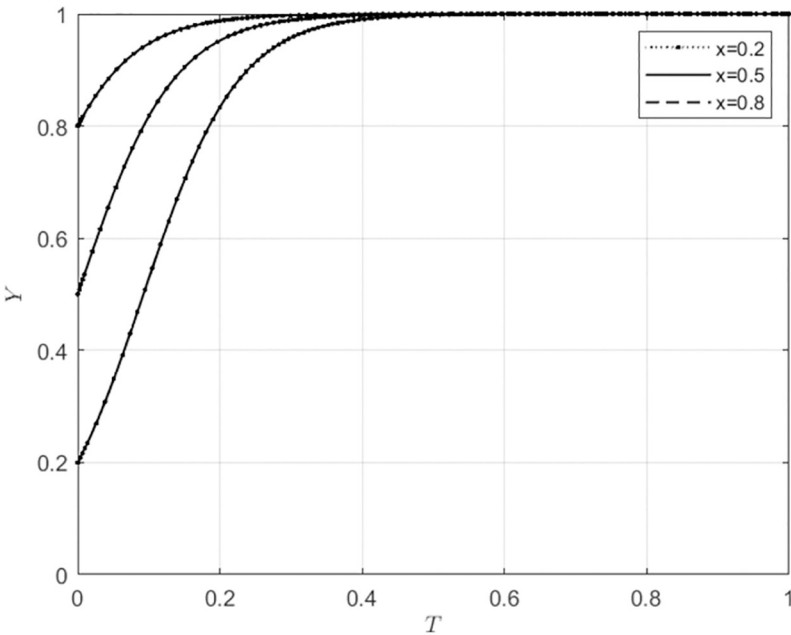

**Fig 4. Local government behavior strategy selection under different initial intentions of agricultural wholesale markets.**

in the three-way game system of agricultural wholesale markets, local government, and wholesalers, the role of local government is more inclined to be the main body that exerts influence, and the behavior choices of agricultural wholesale markets and wholesalers will not affect the evolutionary process of local government behavior choices.

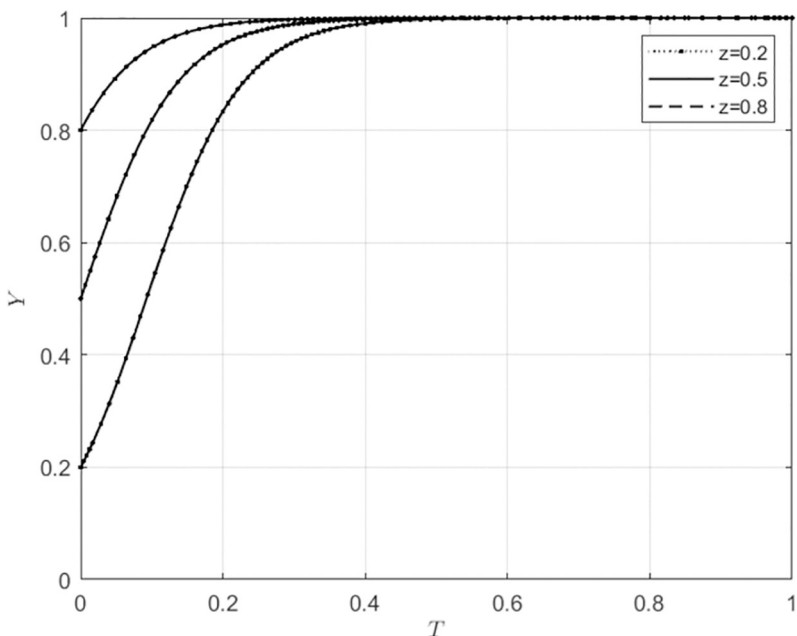

**Fig 5. Local government behavior strategy selection under different initial intentions of wholesalers.**

**5.2.3 Influence of changes in initial intentions of subjects on the evolutionary strategy of wholesalers' behaviors.** The choice of strategy by the subject has different degrees of influence on the strategy selection of wholesalers' behaviors. From Fig 6, it can be observed that when wholesalers' initial intention to choose self-built e-commerce platforms for agricultural wholesale markets is low ($z_0 = 0.2$), indicating a high initial intention to choose third-party e-commerce platforms, wholesalers will choose to use third-party e-commerce platforms if the initial intention of the agricultural wholesale markets to choose self-built e-commerce platforms is also low ($x_0 \leq 0.5$). However, suppose the initial intention of the agricultural wholesale markets to choose self-built e-commerce platforms is high ($x_0 = 0.8$). Wholesalers will use the self-built e-commerce platforms. When wholesalers have a higher initial intention to use the agricultural wholesale markets' self-built e-commerce platforms ($z_0 \geq 0.5$), the speed at which agricultural wholesale markets choose to build their own e-commerce platforms will gradually increase as the initial intention increases. From Fig 7, it can be observed that when wholesalers have a low initial intention to choose to use the agricultural wholesale markets' self-built e-commerce platforms ($z_0 = 0.2$),that is to say, a high intention to choose third-party e-commerce platforms, they will choose the latter regardless of the local government's initial intention to provide subsidies. In the case where wholesalers have a higher initial intention to choose to use the agricultural wholesale markets' self-built e-commerce platforms ($z_0 \geq 0.5$), the size of the subsidies provided by the local government does not affect wholesalers' strategy selection to use the agricultural wholesale markets' self-built e-commerce platforms. However, as the local government's initial intention to provide subsidies gradually increases, the speed at which wholesalers choose to use the self-built e-commerce platforms of agricultural wholesale markets will increase gradually. This indicates that the choices of agricultural wholesale markets and local government can influence the choices of wholesalers, but the choices made by wholesalers tend to have strong inertia. Once wholesalers have built their usage habits, it is difficult for changes in the strategies of other agents to influence the behavior choices of wholesalers.

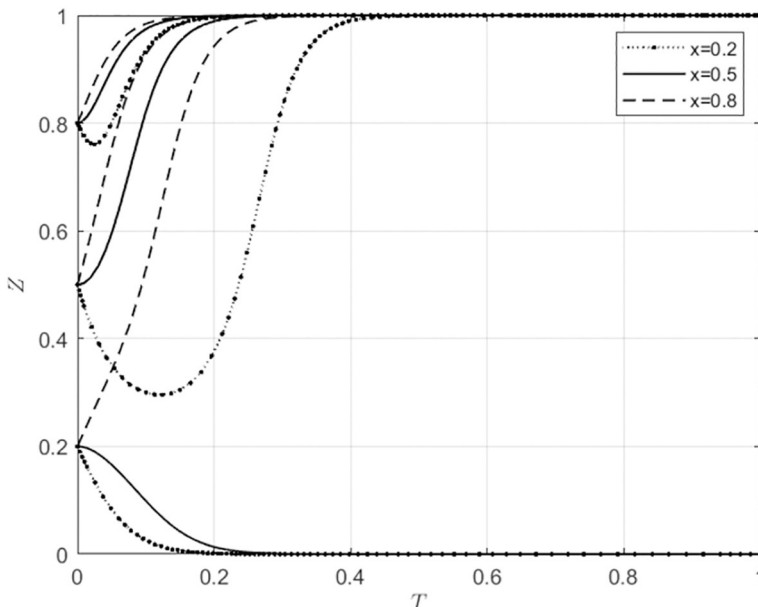

**Fig 6. The behavioral strategy choices of wholesalers in agricultural wholesale markets at different levels of initial intention.**

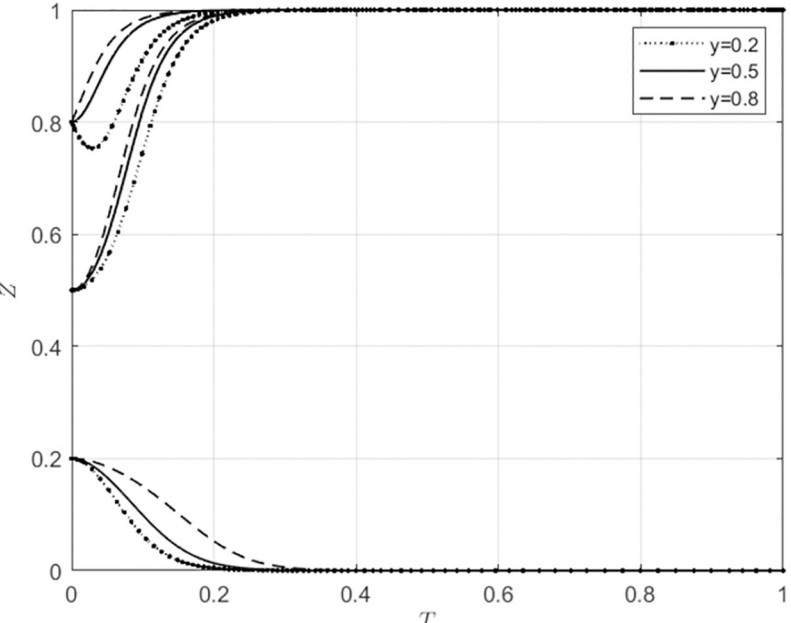

**Fig 7. The behavioral strategy choices of wholesalers in local government at different levels of initial intention.**

## 5.3 The impact of parameter changes on the system evolution results at different levels of the initial intention of the subjects

Based on the above analysis, the values of u, m, b, and Up are individually modified to obtain the strategic evolution trajectory of agricultural wholesale markets and wholesalers with different initial intentions. The above analysis shows that the local government is the main entity exerting influence. Therefore, this study focuses on analyzing the impact of parameter changes on the behavioral choices of agricultural wholesale markets and wholesalers.

**5.3.1 The impact of changes in u on the system evolution outcomes with different initial intention levels of the main subject.**    For $u \in \Omega(0.2, 0.5, 0.8)$, corresponding to low, medium, and high levels of direct subsidies provided by the local government for the agricultural wholesale markets to choose to build their own e-commerce platforms, the evolutionary trajectories of the behaviors of the agricultural wholesale markets and wholesalers are shown in Figs 8 and 9.

According to Fig 8,When the initial willingness of agricultural wholesale markets to build their own e-commerce platforms is low ($x_0 = 0.2$), if the local government provides low-level direct subsidies ($u = 0.2$) to those markets that opt for self-building, they will choose to utilize third-party platforms. Conversely, if the local government offers high-level direct subsidies ($u \geq 0.5$), the agricultural wholesale markets will choose to construct their own e-commerce platforms. Additionally, the pace at which these markets choose to build their own platforms will gradually increase as the local government continues to provide direct subsidies. According to Fig 9, when wholesalers have a low initial intention to use agricultural wholesale markets' e-commerce platforms ($z_0 = 0.2$), if the local government provides a low level of direct subsidies to agricultural wholesale markets for choosing to build their own e-commerce platforms ($u \leq 0.5$), wholesalers will choose to use third-party platforms. However, suppose the local government provides high direct subsidies to agricultural wholesale markets for choosing

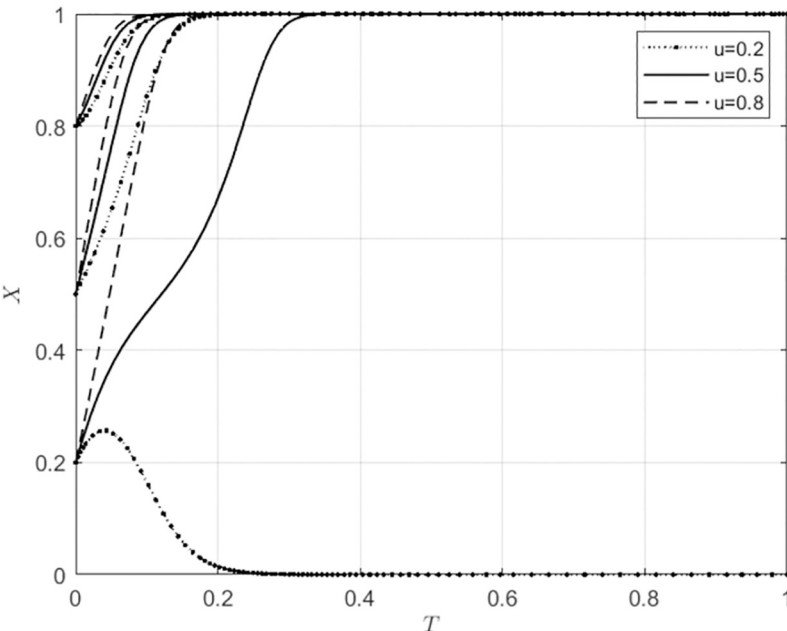

**Fig 8. The impact of u changes on the behavioral choices of agricultural wholesale markets.**

to build their own e-commerce platforms ($u = 0.8$). In that case, wholesalers will use the e-commerce platforms of agricultural wholesale markets. When wholesalers have a moderate initial intention to use agricultural wholesale markets' e-commerce platforms ($z_0 = 0.5$), the direct subsidies provided by the local government to the agricultural wholesale markets for

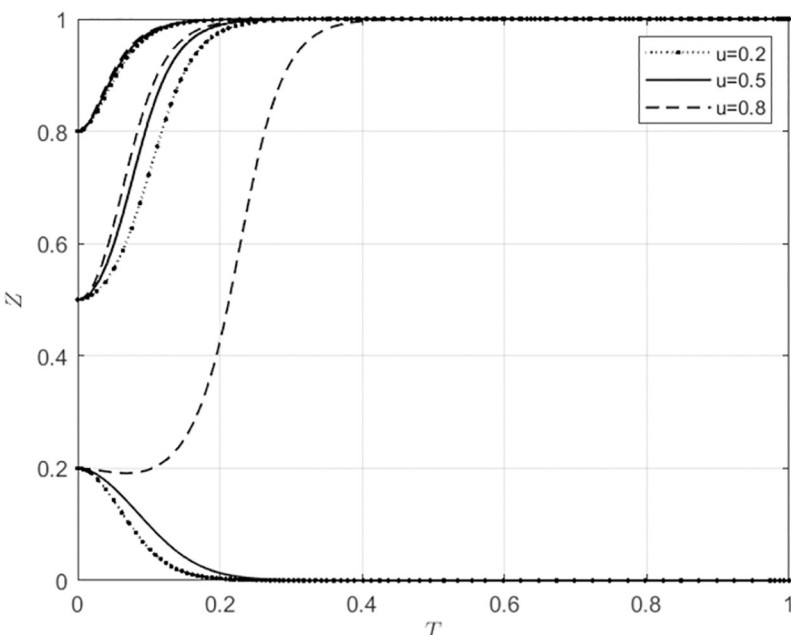

**Fig 9. The impact of u changes on the behavioral choices of wholesalers.**

choosing to build their own e-commerce platforms do not affect wholesalers' behavioral choices, and they will still choose to use the agricultural wholesale markets' e-commerce platforms. However, as the direct subsidies from the local government to the agricultural wholesale markets increase, wholesalers are gradually choosing to use the agricultural wholesale markets' e-commerce platforms at a faster rate. When the initial willingness of wholesalers to use agricultural wholesale markets' e-commerce platforms is high ($x_0$ = 0.8), the direct subsidies provided by the local government for choosing to build their own e-commerce platforms will not affect wholesalers' behavioral choices. Still, it will have a relatively small impact on the development rate using the agricultural wholesale markets' e-commerce platforms. Comparing Figs 8 and 9, it can be seen that when the willingness of the agricultural wholesale markets to choose their e-commerce platforms is high ($x_0 \geq 0.5$), the construction of their own e-commerce platforms can be promoted through direct subsidies provided to the markets for choosing to build their own e-commerce platforms. This also encourages wholesalers to use the e-commerce platforms of agricultural wholesale markets. In cases where the willingness of agricultural wholesale markets to choose their own e-commerce platforms is low ($x_0$ = 0.2), but the willingness to use third-party platforms is high, local government can provide subsidies directly to these markets to encourage the use of third-party e-commerce platforms. This approach is also beneficial for wholesalers who want to use third-party e-commerce platforms offered by agricultural wholesale markets.

**5.3.2 The impact of changes in 1-m on the system evolution outcomes with different initial intention levels of the main subject.** For $1 - m \in \Omega$(0.2, 0.5, 0.8), corresponding to low, medium, and high levels of indirect subsidies provided by the local government to agricultural wholesale markets for choosing to build their own e-commerce platforms, the behavioral strategy evolution trajectories of agricultural wholesale markets and wholesalers can be seen in Figs 10 and 11.

From Fig 10, it can be seen that when the initial willingness of agricultural wholesale markets to build their own e-commerce platforms is low ($x_0$ = 0.2), and if the local government provides a low level of indirect subsidy to agricultural wholesale markets ($1 - m$ = 0.2) to choose to build their own e-commerce platforms, then the agricultural wholesale markets will choose to use third-party e-commerce platforms. On the other hand, if the local government provides a higher level of indirect subsidies ($1 - m \geq 0.5$), the agricultural wholesale markets will choose to build their own e-commerce platforms. When the initial willingness of agricultural wholesale markets to build their own e-commerce platforms is high ($x_0 \geq 0.5$), the behavior of the local government does not affect the choice of agricultural wholesale markets to build their own e-commerce platforms. However, as the level of indirect subsidies the local government provides to agricultural wholesale markets for building their own e-commerce platforms increases, the speed of agricultural wholesale markets building their own e-commerce platforms accelerates gradually. From Fig 11, it can be seen that when the initial willingness of wholesalers to use agricultural wholesale markets' self-built e-commerce platforms is low ($x_0$ = 0.2), and if the local government provides a high level of indirect subsidies to agricultural wholesale markets ($1 - m$ = 0.8) for choosing to build their own e-commerce platforms, then wholesalers will choose to use agricultural wholesale markets' self-built e-commerce platforms. On the other hand, when the initial willingness of wholesalers to use agricultural wholesale markets' self-built e-commerce platforms is high ($x_0 \geq 0.5$), the behavior of the local government does not affect the choice of wholesalers to use the self-built e-commerce platforms of agricultural wholesale markets. However, as the level of indirect subsidies the local government provides to agricultural wholesale markets for building their own e-commerce platforms increases, the rate at which wholesalers choose to use the self-built e-commerce platforms of agricultural wholesale markets also accelerates. Figs 10 and 11, it can be seen that

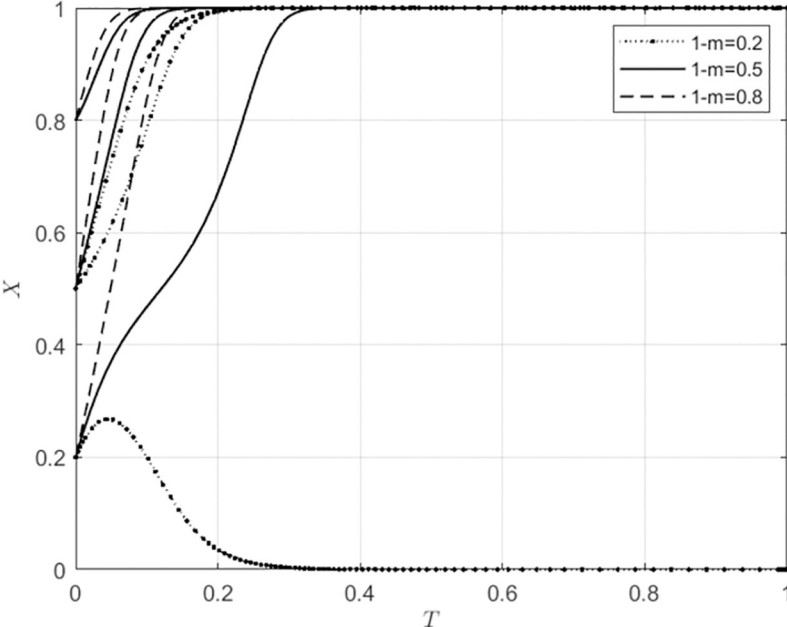

**Fig 10. The impact of 1-m variations on the behavioral choices of agricultural wholesale markets.**

when the agricultural wholesale markets are highly willing to build their own e-commerce platforms, the local government can promote the building of self-built e-commerce platforms by providing indirect subsidies to the agricultural wholesale markets. This indirect subsidies is also beneficial for wholesalers to use the self-built e-commerce platforms of agricultural

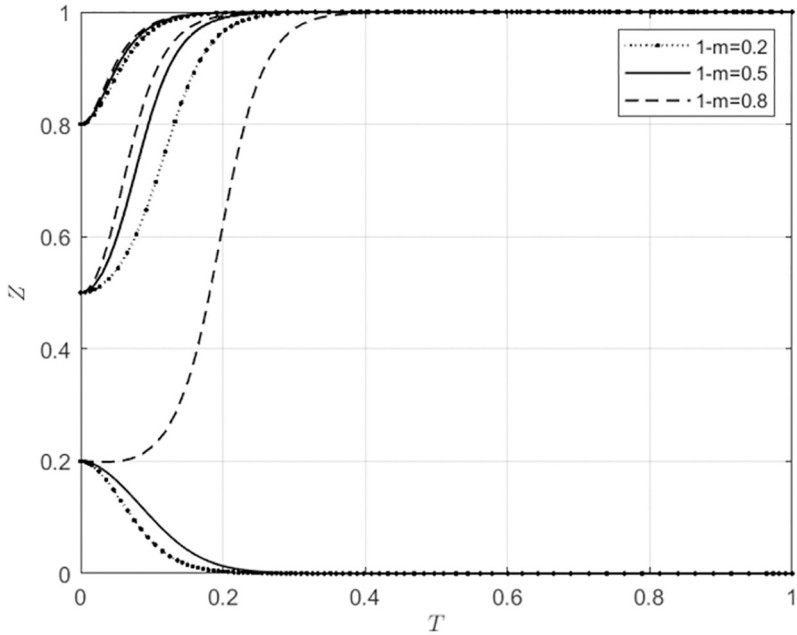

**Fig 11. The impact of 1-m variations on the behavioral choices of wholesalers.**

wholesale markets. Similarly, when the willingness of agricultural wholesale markets to use third-party e-commerce platforms is high, the local government can promote this behavior by providing indirect subsidies to the agricultural wholesale markets; this also helps wholesalers choose to use third-party e-commerce platforms provided by agricultural wholesale markets.

**5.3.3 The impact of changes in b on the system evolution outcomes with different initial intention levels of the main subject.** For $b \in \Omega(30, 60, 90)$, corresponding to low, medium, and high levels of synergistic benefits in the matching of agricultural wholesale markets and wholesalers' behaviors, the evolutionary trajectories of the markets and wholesalers' strategies are shown in Figs 12 and 13.

From Fig 12, it can be observed that when the initial willingness of agricultural wholesale markets to choose self-built e-commerce platforms is low ($x_0 = 0.2$), as the synergistic benefits continue to increase, agricultural wholesale markets opt for third-party e-commerce platforms due to the increasing synergistic benefits. When the initial willingness of agricultural wholesale markets to choose self-built e-commerce platforms is relatively high ($x_0 \geq 0.5$), they will opt to build their own platforms. As the synergistic benefits continue to increase, the speed at which agricultural wholesale markets choose to build their own e-commerce platforms gradually accelerates. From Fig 13, it can be seen that when the initial willingness of wholesalers to use self-built e-commerce platforms of agricultural wholesale markets is low ($x_0 = 0.2$), they will choose to use third-party e-commerce platforms. With the continuous increase of synergistic benefits, the speed at which wholesalers are using third-party e-commerce platforms is gradually increasing. When wholesalers have a high initial willingness ($x_0 \geq 0.5$) to use self-built e-commerce platforms of agricultural wholesale markets, they will utilize these platforms. With the continuous increase in synergistic benefits, the speed of wholesalers using self-built e-commerce platforms of agricultural wholesale markets increases. From Figs 12 and 13, it can be seen that there is a certain degree of synchronization in the behavioral choices of agricultural wholesale markets and wholesalers. With the continuous increase of synergistic benefits, the

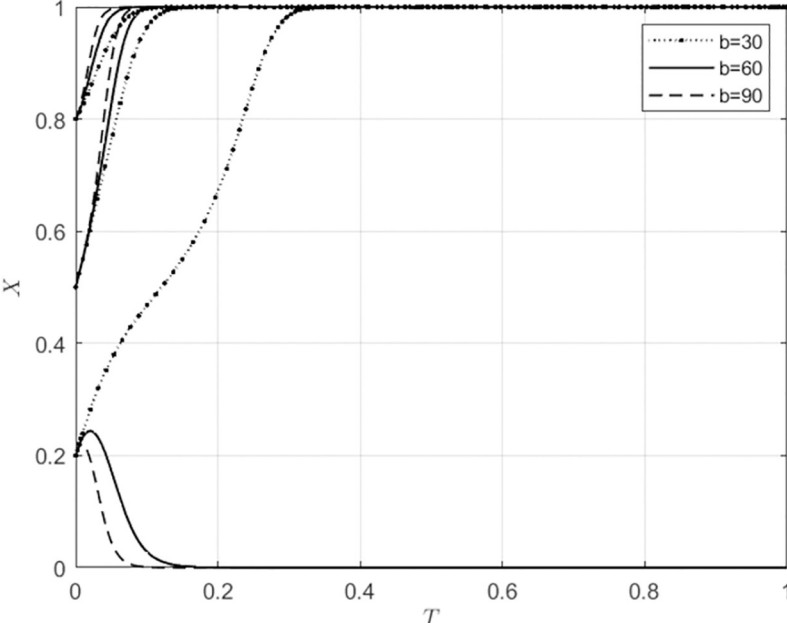

**Fig 12. The impact of b variations on the behavioral choices of agricultural wholesale markets.**

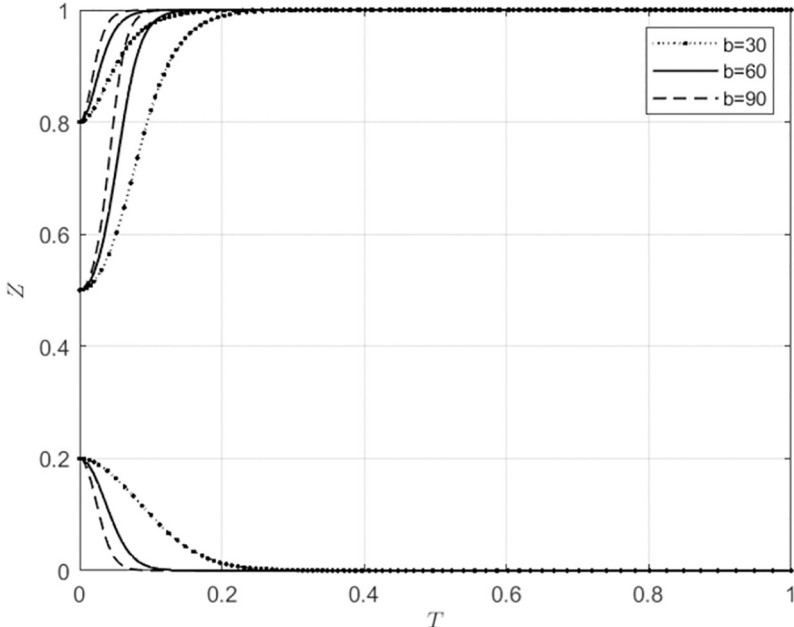

**Fig 13. The impact of b variations on the behavioral choices of wholesalers.**

synchronization of the evolution of behavioral strategies of both parties gradually increases. When agricultural wholesale markets initially show a high willingness to use third-party e-commerce platforms, the presence of higher synergistic benefits can further encourage them to choose to use these platforms. Additionally, wholesalers are more likely to opt for third-party e-commerce platforms. When agricultural wholesale markets have a high initial willingness to build their own e-commerce platforms, with the continuous increase of synergistic benefits, agricultural wholesale markets will choose to build their own e-commerce platforms, and wholesalers will also choose to utilize the self-built e-commerce platforms of agricultural wholesale markets. This indicates that when agricultural wholesale markets choose to build their own e-commerce platforms, they will be more proactive in understanding the needs of wholesalers willing to use their e-commerce platforms. Additionally, wholesalers are more likely to use the self-built e-commerce platforms of agricultural wholesale markets. When agricultural wholesale markets choose to use third-party e-commerce platforms, they will also be more proactive in understanding the needs of wholesalers who are willing to use such platforms. Additionally, wholesalers are more likely to use third-party e-commerce platforms.

**5.3.4 The impact of changes in Up on the system evolution outcomes with different initial intention levels of the main subject.** For $U_p \in \Omega$ (5, 10, 15), corresponding to low, medium, and high levels of additional profits that wholesalers can obtain when they use self-built e-commerce platforms for agricultural wholesale markets. At this time, the evolutionary trajectories of the behavioral strategies for agricultural wholesale markets and wholesalers are shown in Figs 14 and 15.

From Fig 14, it can be seen that when the initial intention of agricultural wholesale markets to build their own e-commerce platforms is low ($x_0 \leq 0.5$), the agricultural wholesale markets will choose to build their own e-commerce platforms. As the additional profits that wholesalers can obtain when using the agricultural wholesale markets' self-built e-commerce platforms continue to increase, the speed at which agricultural wholesale markets choose to build their

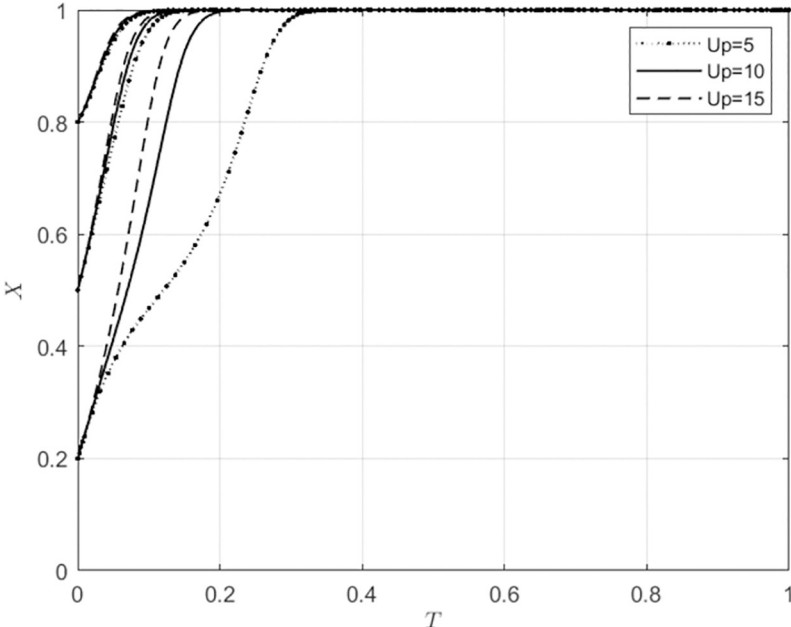

**Fig 14. The impact of Up changes on the behavioral choices of agricultural wholesale markets.**

own platforms accelerates. When the willingness of agricultural wholesale markets to build their own e-commerce platforms is high ($x_0 = 0.8$), the size of the synergistic benefits that wholesalers can obtain when using the agricultural wholesale markets' self-built e-commerce platforms has little impact on the speed at which agricultural wholesale markets choose to

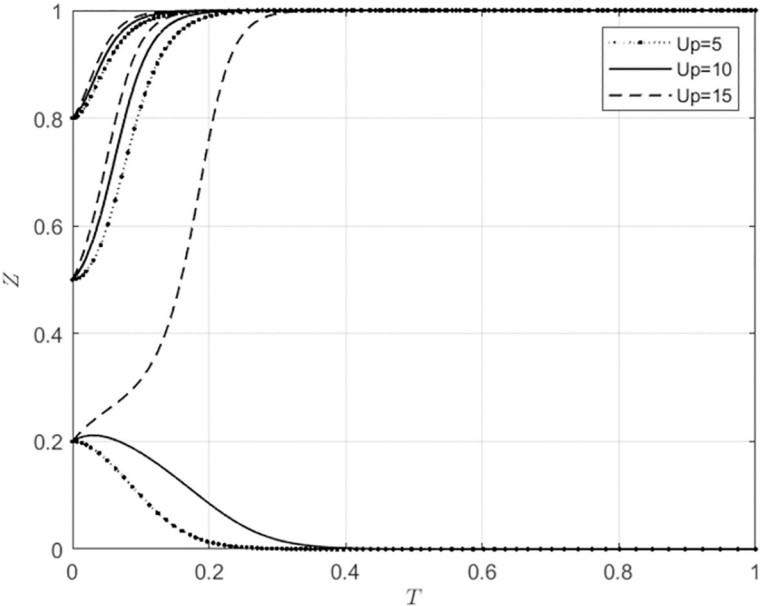

**Fig 15. The impact of Up changes on the behavioral choices of wholesalers.**

build their own e-commerce platforms. From Fig 15, it can be seen that when the initial willingness of wholesalers to use the agricultural wholesale markets' self-built e-commerce platforms is low ($z_0 = 0.2$), if the additional profits that wholesalers can obtain when using the agricultural wholesale markets' self-built e-commerce platforms are low ($U_p \leq 10$), wholesalers will choose to use third-party platforms. Suppose wholesalers can obtain additional profits when using the self-built e-commerce platforms of agricultural wholesale markets, which are highly profitable ($U_p = 15$). In that case, wholesalers will utilize the self-built e-commerce platforms of agricultural wholesale markets. When wholesalers initially show a high willingness ($z_0 \geq 0.5$) to use the self-built e-commerce platforms of agricultural wholesale markets, the speed at which they adopt these platforms gradually increases as the additional profits that wholesalers can obtain increase. This indicates that when wholesalers intend to use the agricultural wholesale markets' self-built e-commerce platforms, the agricultural wholesale markets should strive to increase the additional benefits that wholesalers can obtain when using these platforms. This will encourage wholesalers to utilize the agricultural wholesale markets' self-built e-commerce platforms.

## 6. Discussion

Comparing the stability analysis and simulation results of the game model, it is found that the agricultural wholesale markets, local government, and wholesalers consistently maintain stable strategies in the simulation analysis. During the simulation process, the influence of the initial intentions of the agricultural wholesale markets, local government, and wholesalers on each other's behavioral choices is considered. Variables such as direct subsidies, indirect subsidies, synergistic benefits, and additional profits are introduced. Due to the influence of environmental and decision-making uncertainties on the strategic evolution of the three parties, the decision-making intentions fluctuate randomly. However, the overall trend of the simulation results is consistent with the stability analysis results, demonstrating the validity of this research method.

Meanwhile, the evolutionary results of the game system also demonstrate that the widespread application of digital technologies (artificial intelligence, blockchain, cloud computing, and digital platforms) has provided corresponding technological conditions for the construction of agricultural e-commerce platforms in wholesale markets. However, there are still problems with insufficient funding [51], inadequate policies [52], and difficulty in changing purchasing and sales habits among wholesalers [53] during the construction process of these platforms. Therefore, the use of evolutionary game models and simulations is effective.

## 7. Conclusion

### 7.1 Research conclusions

1. In the game system comprising agricultural wholesale markets, local government, and wholesalers, the behavioral choices of agricultural wholesale markets are influenced by the initial intentions of the local government and wholesalers. When agricultural wholesale markets have a higher initial intention to choose self-built e-commerce platforms, increased subsidies from the local government and higher intentions of wholesalers to use agricultural wholesale markets' self-built e-commerce platforms are conducive to promoting the use of self-built e-commerce platforms in agricultural wholesale markets. When agricultural wholesale markets show a higher intention to choose third-party e-commerce platforms, increased subsidies from the local government and a stronger intention among wholesalers to utilize these platforms can be beneficial for agricultural wholesale markets to use third-party e-commerce platforms. The local government is the main entity exerting influence,

and the behavioral choices of agricultural wholesale markets and wholesalers can not change the behavioral choices of the local government. The behavioral choices of agricultural wholesale markets and the local government influence the behavioral choices of wholesalers. However, the behavioral choices of wholesalers tend to exhibit strong inertia. Once they form their usage habits, it becomes challenging for external entities to alter their behavioral choices.

2. The behavioral choices of agricultural wholesale markets and wholesalers have a certain level of synchronization. As the mutual benefits of their actions increase, the synchronicity of their behavioral strategies is also enhanced. If agricultural wholesale markets have a strong preference for creating their own e-commerce platforms, providing direct subsidies from the local government to these markets will encourage the development of such platforms. These subsidies will also encourage wholesalers to use the self-built e-commerce platforms of agricultural wholesale markets. Indirect subsidies from the local government to agricultural wholesale markets that choose self-built e-commerce platforms also encourage the adoption of these platforms and facilitate wholesalers in their decision to use them. The gradual increase in additional benefits for wholesalers who utilize self-built e-commerce platforms in agricultural wholesale markets is beneficial for promoting these options.

3. When agricultural wholesale markets initially show a strong preference for third-party e-commerce platforms, direct subsidies from the local government can encourage the adoption of third-party e-commerce platforms by agricultural wholesale markets. Additionally, these subsidies also benefit the promotion of wholesalers' choice of third-party e-commerce platforms. Indirect subsidies from the local government to agricultural wholesale markets that choose third-party e-commerce platforms also encourage the adoption of these platforms by agricultural wholesale markets and facilitate the use of third-party e-commerce platforms by wholesalers.

## 7.2 Policy recommendations

1. As an important policy maker, the government should take proactive measures to promote the development of the agricultural wholesale markets through e-commerce. In recent years, the trend of third-party e-commerce platforms' monopoly has become increasingly apparent. These platforms not only charge high usage fees and commissions but also impose restrictions on wholesalers. This situation has prompted eligible companies to transition to the self-built e-commerce platforms gradually. In addition, the "14th Five-Year Plan for Digital Economy Development" issued by the State Council in 2022 also mentioned supporting large enterprises with certain conditions to establish comprehensive e-commerce platforms. Therefore, if enterprises have a strong interest in building their own e-commerce platforms, the government can provide funding through fiscal budget arrangements to subsidize the construction of agricultural wholesale markets that choose to build their own e-commerce platforms to encourage and support the establishment of these platforms. In addition, the government can lower the cost of building self-built e-commerce platforms for agricultural wholesale markets through measures such as monetary supply and interest rate regulation, such as lowering loan interest rates, to further promote the construction of these platforms. The "Notice on Promoting High-Quality Development of E-commerce Measures" issued by the Shandong Provincial People's Government also emphasizes encouraging traditional commercial distribution enterprises to expand their

network sales share through self-built or third-party platforms. When the agricultural wholesale markets choose to use third-party e-commerce platforms, the government can directly subsidize the platform service fees that these markets pay for maintenance, technical support, and promotion. At the same time, the government can help agricultural wholesale markets reduce the resources and human resources required for using third-party e-commerce platforms, as well as the expenses of managing and operating an e-commerce business, by providing low-interest loans and other methods.

2. Agricultural wholesale markets and wholesalers' behavior choices are somewhat synchronized. When agricultural wholesale markets choose to build their own e-commerce platforms, they can gather information about the specific requirements of wholesalers through face-to-face interactions and online research. By utilizing data analysis tools to analyze wholesalers' behavioral data and behavior trajectories, agricultural wholesale markets can gain insights into wholesalers' search and purchase preferences and identify any problems they may encounter when using the platforms. This insight can help agricultural wholesale markets better understand the needs of wholesalers who are willing to use self-built e-commerce platforms in agricultural wholesale marketsï¼? which can promote the use of such platforms by wholesalers. Moreover, agricultural wholesale markets can customize their independent and self-built e-commerce platforms according to the needs and preferences of wholesalers. This allows them to provide better user experiences and enhance their competitiveness in the market,and also makes it easier for wholesalers to browse and purchase products, enjoy personalized services, and obtain accurate product information, thereby enhancing wholesalers' satisfaction and generating additional revenue. When agricultural wholesale markets choose to use third-party platforms, they can actively participate in various activities organized by the platforms to create a tight marketing atmosphere with other merchants. This can promote the use of the platforms by wholesalers who are willing to use third-party e-commerce platforms.

## 7.3 Contributions and limitations

This research makes significant contributions to both theoretical knowledge and management practices. Firstly, the research findings reveal the initial intentions of agricultural wholesale markets, local government, and wholesalers regarding each other's behavioral choices, which helps to deepen the understanding of the strategic decision-making and interaction processes among the different participants in the tripartite evolutionary game environment. Secondly, the research proposes policy recommendations for building e-commerce platforms for agricultural wholesale markets by analyzing government subsidies, collaborative benefits, and additional benefits. In particular, government subsidies and collaborative benefits serve as important influencing factors, and it is recommended that the government increase funding to support and cooperation efforts for agricultural wholesale markets to promote their development towards e-commerce. Furthermore, while it may be challenging to attain additional benefits directly, it is advisable for agricultural wholesale markets to enhance their internal innovation capabilities, optimize product supply, and improve service quality to enhance market competitiveness and attractiveness. This research provides valuable experiences and insights for transforming agricultural wholesale markets in e-commerce. It is also a reference and foundation for policy formulation and management practices. However, this research also has certain limitations. During the construction of the game model, only the mutual influence between agricultural wholesale markets, local government, and wholesalers was considered. Other aspects, such as logistics during the construction of e-commerce platforms for

agricultural wholesale markets, should have been considered. Therefore, future research can further advance the construction process of e-commerce platforms for agricultural wholesale markets by considering more participating entities, thus promoting the e-commerce transformation process more effectively.

## Supporting information

**S1 File.**
(DOCX)

## Author Contributions

**Formal analysis:** Yongtao Liu.

**Funding acquisition:** Qianwen Luo.

**Methodology:** Qianwen Luo, Yujie Wang.

**Software:** Yujie Wang.

**Visualization:** Yongtao Liu.

**Writing – original draft:** Yujie Wang.

**Writing – review & editing:** Qianwen Luo.

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
