## [Decision Letter · Decision Letter 0]

24 Oct 2023

PONE-D-23-31750Research on the Construction and Promotion Strategies of E-commerce Platforms in Agricultural Wholesale MarketsPLOS ONE

Dear Dr. Luo,

Thank you for submitting your manuscript to PLOS ONE. After careful consideration, we feel that it has merit but does not fully meet PLOS ONE’s publication criteria as it currently stands. Therefore, we invite you to submit a revised version of the manuscript that addresses the points raised during the review process.

We look forward to receiving your revised manuscript.

Kind regards,

Jitendra Yadav, Ph.D.

Academic Editor

PLOS ONE

Journal Requirements:

   "Beijing Municipal Social Science Foundation Project Grant, grant number 2020ZX028,Beijing Municipal Education Commission Scientific Research Programme Project Grant,grant number SM202310037004"

3. Please include your tables as part of your main manuscript and remove the individual files. Please note that supplementary tables (should remain/ be uploaded) as separate "supporting information" files

Additional Editor Comments:

As observed by the reviewers, the manuscript in the current form suffers from variety of lacunas. In addition to the observations of the reviewers, I believe the following sections need to be further developed:

1. The introduction section needs to provide more insight and primarily resolve the three key questions namely (a) what is the need of this research, (b) what are the research gaps identified by the authors, and (c) what is contribution of this manuscript in reference to academia and practice.

2. The literature review segment can be further enhanced by creating two sub-headings namely (a) State of Agriculture Wholesale Market, and (b) promotional strategies of e-commerce platforms.

The subheadings can be altered as per the understanding of the authors but the object is to provide a robust insightful literature review of the domain. Some of the suggested studies that author(s) can include and refer while re-drafting the manuscript:

a) https://doi.org/10.1504/IJBC.2020.108997

b) https://doi.org/10.1142/9789811205279_0005

c) https://doi.org/10.17705/1CAIS.05240

3. The quality of images is poor. Kindly enhance the quality of the images.

4. As observed by one of the reviewers, enhance the discussion section.

5. The referencing style is incorrect. Authors should adhere to the journal formatting guidelines and update the referencing style.

Reviewers' comments:

Reviewer's Responses to Questions

**Comments to the Author**

1. Is the manuscript technically sound, and do the data support the conclusions?

Reviewer #1: Partly

Reviewer #2: Partly

2. Has the statistical analysis been performed appropriately and rigorously? 

Reviewer #1: No

Reviewer #2: No

3. Have the authors made all data underlying the findings in their manuscript fully available?

Reviewer #1: Yes

Reviewer #2: No

4. Is the manuscript presented in an intelligible fashion and written in standard English?

Reviewer #1: Yes

Reviewer #2: Yes

5. Review Comments to the Author

Reviewer #1: The reviewer believes that the topic “Research on the Construction and Promotion Strategies of E-commerce Platforms in Agricultural Wholesale Markets” is worthy of investigation. However, the following needs to be addressed. There are minor and major issues that should be corrected. I believe the paper could be further strengthened by added information about.

1.The title does not provide a core theme of the topic.

2.Please specify the source of the simulation data.

3.The language of this manuscript needs help from native speakers.

4.Please underscore the scientific value-added to your paper in your abstract. Your abstract should clearly state the essence of the problem you are addressing, what you did and what you found and recommend. That would help a prospective reader of the abstract to decide if they wish to read the entire article.

5.LINES31-49. This a very vague statement. These sentences do not provide any information on how the concept could be conceptualized? - The Introduction should have 1) a concise but complete justification of the topic's importance both academically and practically, and 2) an explanation of the gaps both in research and practice. Please review appropriate literature in the Introduction, with the research question clearly arising from that review.

6.LINES42-48.What authors wanted to convey. Here author must build research gap following the previous studies.-The manuscript does not answer the following concerns: Why is it timeliness to explore such a study? What makes this study different from the previously published studies? Are there any similarly findings in line with the previously published studies? Are the findings different from prior academic studies that were conducted elsewhere, if any? What it requires, what are the new technologies, some recent issue highlights the importance. See the following: A three evolutionary game model for driving mechanism of industry-university-research collaborative innovation in agricultural innovation ecosystems. Plos one, 18(8), e0289408.；

New Energy-Driven Construction Industry: Digital Green Innovation Investment Project Selection of Photovoltaic Building Materials Enterprises Using an Integrated Fuzzy Decision Approach. Systems 2023, 11, 11. https://doi.org/10.3390/systems11010011.

7.-There is no flow in the text. It partly depends on the lack of proofreading but also on the fact that many statements and claims are made without being followed up by a clear and logical discussion. It is especially problematic in the Introduction that brings up a number of findings from different areas without linking them together.

8.-More importantly, the choice of the questionnaire questions should be explained in light of the theory and the prior literature on the topic. The arguments are simply relationships and causes very close to the replication of many studies dealing with the same thing. For example, what is connection of upgrading path of manufacturing enterprises from the value perspective. See the following: Developing a conceptual partner matching framework for digital green innovation of agricultural high-end equipment manufacturing system toward agriculture 5.0: A Novel Niche Field Model Combined With Fuzzy VIKOR. Frontiers in Psychology, 13, 924109.

9.-Methodology: Model.. I suggest authors here build your main heading on Research and data methodology. Clearly explain the model building process, and what previous studies have used similar models (model testing approach).

See the following: "Incentive Mechanism for the Development of Rural New Energy Industry: New Energy Enterprise–Village Collective Linkages considering the Quantum Entanglement and Benefit Relationship", International Journal of Energy Research. https://doi.org/10.1155/2023/1675858.

A stochastic differential game of low carbon technology sharing in collaborative innovation system of superior enterprises and inferior enterprises under uncertain environment, https://doi.org/10.1515/math-2018-0056

10.The authors should emphasize the important role of digital technology in future research. See the following: The Interaction Mechanism and Dynamic Evolution of Digital Green Innovation in the Integrated Green Building Supply Chain. Systems 2023, 11, 122. https://doi.org/10.3390/systems11030122.

11.Please consider this structure for manuscript final part.

-Discussion

-Conclusion

-Managerial Implication

-Practical/Social Implications

12.Please make sure your conclusions' section underscores the scientific value-added of your paper, and/or the applicability of your findings/results. Highlight the novelty of your study. In addition to summarizing the actions taken and results, please strengthen the explanation of their significance. It is recommended to use quantitative reasoning comparing with appropriate benchmarks, especially those stemming from previous work. .

Reviewer #2: This paper employs a three-party game model involving the agricultural wholesale markets, the government, and traders to analyze the interests and behavioral choices of each party. The research findings indicate that government subsidies play a key role in the establishment of ecommerce platforms in agricultural wholesale markets. Although the content of this manuscript is relatively rich, but I think the problems and some shortcomings are as follows:

1. In line 8-11, the Abstract points out that “The research findings indicate that government subsidies play a key role in the establishment of e-commerce platforms in agricultural wholesale markets. When government subsidies are sufficient, the agricultural wholesale markets are more willing to choose the establishment of e-commerce platforms.” In reality, agricultural products are mostly sold on various e-commerce platforms, and the proportion of self-established e-commerce platforms is not high. The wholesale market for agricultural products mainly exists offline, and its main role is to provide a market for agricultural product trading. The establishment of agricultural product e-commerce platforms focuses more on the operation and maintenance of e-commerce platforms. The durability of government subsidies is an important consideration.

2. In Introduction, this paper fails to elaborate on the necessity of building a self-built e-commerce platform for agricultural product wholesale markets. which is not convincing.

3. In literature review, rarely discusses the application of system dynamics principles in the selection of government subsidy strategies.

4. In Game process analysis, line 129-131, this paper assumed that “In order to encourage the agricultural wholesale markets to establish e-commerce platforms, the government will provide subsidies to the markets”. In reality, more professional platform providers establish and operate e-commerce platforms than wholesale markets for agricultural products. This assumption seems to be inconsistent with reality.

5. In Parameter settings, line 158-160, this paper argued that “The additional profits gained from establishing e-commerce platforms include benefits derived from the platforms’ advantages in quick supply-demand matching, information transmission, and capital support.” But this is not the case in reality. The cultivation and promotion of any e-commerce platform may incur huge sunk costs.

6. In Parameter settings, line 172-173, this paper pointed out that “The government’s adoption of subsidy measures can enhance confidence and generate positive social reputation effects, denoted as A.” How to measure A?

7. The biggest problem in constructing Payoff Matrix based on system dynamics principles is relative idealization, which makes it difficult to consider multiple factors. This article fails to solve this problem.

8. The Payoff Matrix constructed in the paper has several possibilities, but it is only a theoretical comparison. The practical significance is relatively small.

9. In practice, the existence and form of government subsidies are not key factors in establishing e-commerce platforms for agricultural wholesale markets.

10. All formulas are not numbered.

11. What is the basis for parameter assignment in Analysis of the strategy?

6. PLOS authors have the option to publish the peer review history of their article (what does this mean?). If published, this will include your full peer review and any attached files.

Reviewer #1: No

Reviewer #2: No

---

## [Author Response · Author response to Decision Letter 0]

25 Nov 2023

We provided detailed answers to each question in our “Response to Reviewers” letter.

---

## [Decision Letter · Decision Letter 1]

12 Dec 2023

PONE-D-23-31750R1Strategies selection for building e-commerce platforms for agricultural wholesale markets:A tripartite evolutionary game perspectivePLOS ONE

Dear Dr. Luo,

Thank you for submitting your manuscript to PLOS ONE. After careful consideration, we feel that it has merit but does not fully meet PLOS ONE’s publication criteria as it currently stands. Therefore, we invite you to submit a revised version of the manuscript that addresses the points raised during the review process.

Thank you for submitting your manuscript to our journal. After a thorough review process, the reviewers have provided constructive feedback, indicating the need for a major revision before the manuscript can be considered for publication. The reviewers have highlighted several key concerns. Addressing these issues is crucial for enhancing the overall quality and credibility of your work. To ensure a successful revision, please carefully consider and respond to each reviewer comment. Clearly outline the modifications made in response to their suggestions and provide a detailed explanation if any comment is not addressed.

We look forward to receiving your revised manuscript.

Kind regards,

Jitendra Yadav, Ph.D.

Academic Editor

PLOS ONE

Reviewers' comments:

Reviewer's Responses to Questions

**Comments to the Author**

1. If the authors have adequately addressed your comments raised in a previous round of review and you feel that this manuscript is now acceptable for publication, you may indicate that here to bypass the “Comments to the Author” section, enter your conflict of interest statement in the “Confidential to Editor” section, and submit your "Accept" recommendation.

Reviewer #1: (No Response)

Reviewer #2: (No Response)

2. Is the manuscript technically sound, and do the data support the conclusions?

Reviewer #1: (No Response)

Reviewer #2: Partly

3. Has the statistical analysis been performed appropriately and rigorously? 

Reviewer #1: (No Response)

Reviewer #2: Yes

4. Have the authors made all data underlying the findings in their manuscript fully available?

Reviewer #1: (No Response)

Reviewer #2: Yes

5. Is the manuscript presented in an intelligible fashion and written in standard English?

Reviewer #1: (No Response)

Reviewer #2: Yes

6. Review Comments to the Author

Reviewer #1: The content is supplemented according to the article and method suggestions provided by the reviewers. After summarizing the shortcomings of previous studies, the theoretical framework and model building process of this paper are introduced, and the timeliness and novelty of this paper are compared. You ignored my comments. The authors ignored a lot of the literature in my previous comments, which is very disrespectful. If the authors can't solve this problem, I have to reject the manuscript.

Please explore this issue. In the revised paper, compared with the following literatures, what is the novelty of this study?

See the following: "Incentive Mechanism for the Development of Rural New Energy Industry: New Energy Enterprise–Village Collective Linkages considering the Quantum Entanglement and Benefit Relationship", International Journal of Energy Research. https://doi.org/10.1155/2023/1675858.

A stochastic differential game of low carbon technology sharing in collaborative innovation system of superior enterprises and inferior enterprises under uncertain environment, https://doi.org/10.1515/math-2018-0056

Please integrate the paper and cite the corresponding literature..

Reviewer #2: It is worth encouraging that the author has made a lot of revisions to the manuscript, but there are still some deficiencies in the revised manuscript, such as:

1. For review comments 6, the meaning of “A” is not fully explained.

2. Using the principle of system dynamics to make decisions based on the income matrix can best be combined with the following policy recommendations.

3. According to review comments 11, when using data for simulation verification, the relationship between data is difficult to be convincing.

7. PLOS authors have the option to publish the peer review history of their article (what does this mean?). If published, this will include your full peer review and any attached files.

Reviewer #1: No

Reviewer #2: No

---

## [Author Response · Author response to Decision Letter 1]

19 Dec 2023

Thank you very much for the constructive suggestions from the reviewers. We have addressed each issue raised in the “Response to Reviewers” letter.

---

## [Decision Letter · Decision Letter 2]

4 Jan 2024

Strategies selection for building e-commerce platforms for agricultural wholesale markets:A tripartite evolutionary game perspective

PONE-D-23-31750R2

Dear Dr. Luo,

We’re pleased to inform you that your manuscript has been judged scientifically suitable for publication and will be formally accepted for publication once it meets all outstanding technical requirements.

Kind regards,

Jitendra Yadav, Ph.D.

Academic Editor

PLOS ONE

Reviewers' comments:

Reviewer's Responses to Questions

**Comments to the Author**

1. If the authors have adequately addressed your comments raised in a previous round of review and you feel that this manuscript is now acceptable for publication, you may indicate that here to bypass the “Comments to the Author” section, enter your conflict of interest statement in the “Confidential to Editor” section, and submit your "Accept" recommendation.

Reviewer #1: (No Response)

Reviewer #2: (No Response)

2. Is the manuscript technically sound, and do the data support the conclusions?

Reviewer #1: (No Response)

Reviewer #2: (No Response)

3. Has the statistical analysis been performed appropriately and rigorously? 

Reviewer #1: (No Response)

Reviewer #2: (No Response)

4. Have the authors made all data underlying the findings in their manuscript fully available?

Reviewer #1: (No Response)

Reviewer #2: (No Response)

5. Is the manuscript presented in an intelligible fashion and written in standard English?

Reviewer #1: (No Response)

Reviewer #2: (No Response)

6. Review Comments to the Author

Reviewer #1: The manuscript has significantly improved as compared to the previous version. Indeed, the authors tried to improve it, and the main weaknesses are solved.

Thus, in my opinion, the manuscript is recommendable for publication..

Reviewer #2: (No Response)

7. PLOS authors have the option to publish the peer review history of their article (what does this mean?). If published, this will include your full peer review and any attached files.

Reviewer #1: No

Reviewer #2: No

---

## [Editor Report · Acceptance letter]

17 Jan 2024

PONE-D-23-31750R2 

PLOS ONE

Dear Dr. Luo, 

I'm pleased to inform you that your manuscript has been deemed suitable for publication in PLOS ONE. Congratulations! Your manuscript is now being handed over to our production team.

Kind regards, 

on behalf of

Dr. Jitendra Yadav 

Academic Editor

PLOS ONE